# A Fluorogenic Assay: Analysis of Chemical Modification of Lysine and Arginine to Control Proteolytic Activity of Trypsin

**DOI:** 10.3390/molecules26071975

**Published:** 2021-03-31

**Authors:** Kunal N. More, Tae-Hwan Lim, Julie Kang, Dong-Jo Chang

**Affiliations:** College of Pharmacy and Research Institute of Life and Pharmaceutical Sciences, Sunchon National University, 255 Jungang-ro, Suncheon 57922, Korea; kunalmore83@gmail.com (K.N.M.); c79852er@naver.com (T.-H.L.); juliy19@naver.com (J.K.)

**Keywords:** fluorogenic peptide substrate, trypsin, fluorogenic assay, chemoselective chemical modification of amino acids

## Abstract

The chemical modification of amino acids plays an important role in the modulation of proteins or peptides and has useful applications in the activation and stabilization of enzymes, chemical biology, shotgun proteomics, and the production of peptide-based drugs. Although chemoselective modification of amino acids such as lysine and arginine via the insertion of respective chemical moieties as citraconic anhydride and phenyl glyoxal is important for achieving desired application objectives and has been extensively reported, the extent and chemoselectivity of the chemical modification of specific amino acids using specific chemical agents (blocking or modifying agents) has yet to be sufficiently clarified owing to a lack of suitable assay methodologies. In this study, we examined the utility of a fluorogenic assay method, based on a fluorogenic tripeptide substrate (FP-AA1-AA2-AA3) and the proteolytic enzyme trypsin, in determinations of the extent and chemoselectivity of the chemical modification of lysine or arginine. As substrates, we used two fluorogenic tripeptide probes, MeRho-Lys-Gly-Leu(Ac) (lysine-specific substrate) and MeRho-Arg-Gly-Leu(Ac) (arginine-specific substrate), which were designed, synthesized, and evaluated for chemoselective modification of specific amino acids (lysine and arginine) using the fluorogenic assay. The results are summarized in terms of half-maximal inhibitory concentrations (IC_50_) for the extent of modification and ratios of IC_50_ values (IC_50_arginine/IC_50_lysine and IC_50_lysine/IC_50_arginine) as a measure of the chemoselectivity of chemical modification for amino acids lysine and arginine. This novel fluorogenic assay was found to be rapid, precise, and reproducible for determinations of the extent and chemoselectivity of chemical modification.

## 1. Introduction

Protein modification is important in the fields of chemical biology, shotgun proteomics, and peptide therapeutics [1,2,3,4,5]. Post-translational protein modification (PTM) is a protein modification process that occurs during protein biosynthesis subsequent to translation [2,6] and involves chemical transformations that produce chemically and functionally diverse proteins via the covalent addition of chemical moieties to the amino acid side chains within proteins [7]. Similar modifications can be achieved experimentally by exploiting the vast range of chemical reactions to facilitate the conjugation of amino acids including lysine and arginine with specific chemical moieties [3,8,9]. A large number of studies have been conducted to date, regarding specific modification of lysine with citraconic anhydride [10,11,12,13] and arginine with phenyl glyoxal [14,15,16]. Shapiro et al. have described the methods of chemical modification of lysine and arginine with specific chemical reagents and evaluated the effect of modification on ribonucleolytic activity of angiogenin [17]. Under optimal reaction conditions, chemical reagents such as citraconic anhydride and phenyl glyoxal covalently react with nucleophilic amino or guanidine group of lysine and arginine in substrates, which interrupts the proteolysis of substrates by trypsin. Reported chemical reagents for lysine modification belong to the class of anhydrides (citraconic anhydride, acetic anhydride, and diethylpyrocarbonate) while the arginine modification exploits the chemistry of 1,2-dicarbonyl compounds (phenyl glyoxal and *p*-hydroxy phenyl glyoxal). Although a number of reports have been published regarding the modification of amino acids including lysine and arginine based on the nature of the side chain groups [8,18,19], the extent and chemoselectivity of modification are not well documented, owing to a lack of practical and reliable assay methods. In an attempt to quantify exact extent and chemoselectivity of reported blocking agents for either lysine or arginine, we have planned to design and develop a highly reliable, fast, and practical assay method. Fluorogenic assays are highly sensitive, precise, and rapid, can be used for the determination of enzyme-substrate reactions and fluorogenic peptide substrates are routinely used to determine the activity of proteases such as endopeptidases, carboxypeptidases, and aminopeptidases [20,21,22]. The fluorogenic peptide substrates that have been reported to date consist of a fluorophore-conjugated peptide with a specific sequence for enzyme recognition. The cleavage of a specific peptide bond by proteolytic enzymes as trypsin releases the fluorophore, and the corresponding increase in fluorescence is determined as a measured of protease activity. We speculated that chemical modification (blocking) of an amino acid including lysine and arginine at the proteolytic site would render the peptide substrate resistant to proteolysis by protease enzymes as trypsin and that the fluorescence would decrease corresponding to the extent of chemical modification (Figure 1).

To test our hypothesis, we designed fluorogenic tripeptide substrates containing a lysine or arginine, which serves as a site for cleavage by a specific proteolytic enzyme, trypsin. Within the structure of the fluorogenic peptide substrate, which comprises a fluorophore (FP) attached to a tripeptide (FP-AA1-AA2-AA3), AA1 represents the site (lysine or arginine) for cleavage by trypsin (Figure 2). In accordance with our hypothesis, the reaction of trypsin with the FP-AA1-AA2-AA3 tripeptide would result in an increase in fluorescence intensity, whereas conversely, selective blocking of AA1 with a suitable blocking agent (chemical modifying agents) would inhibit trypsin proteolysis, and there would be a concomitant reduction in fluorescence, depending on the extent of modification. Trypsin is a serine protease that cleaves peptides or proteins selectively at the carboxyl side of the lysine or arginine [5,23]. Thus, we selected lysine or arginine as the AA1 residue attached directly to the fluorophore MeRho, which is a good fluorophore in terms of fluorescence intensity, biocompatibility, and stability [24]. The AA2-AA3 (Gly-Leu(Ac)) sequence was similarly selected based on trypsin substrate specificity [25].

In this study, we synthesized two fluorogenic tripeptide substrates, MeRho-Lys-Gly-Leu(Ac) (lysine-specific substrate) and MeRho-Arg-Gly-Leu(Ac) (arginine-specific substrate), and evaluated the blocking activity of known lysine or arginine blocking agents to assess the extent and chemoselectivity of chemical modification. Concentration–response curves were generated based on the concentration-dependent blocking of AA1 by lysine or arginine blocking agents. As a measure of the extent of chemical modification, we calculated the half-maximal inhibitory concentrations (IC_50_) of AA1 blocking agents from semilogarithmic plots of concentration–response curves. Additionally, the ratios of the IC_50_ values for lysine and arginine (IC_50_arginine/IC_50_lysine and IC_50_lysine/IC_50_arginine) were calculated as a measure of the chemoselectivity of chemical modification.

## 2. Results and Discussion

### 2.1. Design and Synthesis of Fluorogenic Peptide Substrates for Fluorogenic Assay

The proteolytic activity of trypsin is measured based on an increase in fluorescence subsequent to proteolysis of a fluorogenic peptide substrate [26]. The mechanism of action of all protease-targeted fluorogenic substrates is based on an increase in fluorescence promoted via the proteolytic site-selective cleavage of peptide bonds, followed by the release of free fluorophore [27]. We accordingly designed a fluorogenic assay to determine the extent and chemoselectivity of the chemical modification of amino acids lysine and arginine. According to our assumption, blocking of lysine or arginine at peptide cleavage site in the fluorogenic peptide substrate, via the action of a chemical modifying agent (blocking agent), would inhibit proteolysis by trypsin, and thereby prevent fluorescence emission. In addition, the intensity of fluorescence upon cleavage by a trypsin following chemical modification can provide information regarding the extent and chemoselectivity of the modification. For the practical application of our hypothesis, we performed a study using trypsin, that cleaves the carboxyl side of lysine or arginine in the peptide sequence of fluorogenic peptide substrate. As trypsin substrates containing lysine and arginine, we synthesized the conjugated peptides MeRho-Lys-Gly-Leu(Ac) and MeRho-Arg-Gly-Leu(Ac), and evaluated the extent of chemical modification and chemoselectivity for either lysine or arginine using known lysine or arginine blocking agents [8,17].

The fluorogenic tripeptide substrates were synthesized from methylrhodol (MeRho), as depicted in Scheme 1, which entailed a series of amide couplings and deprotections. Initially, compound **1,** with an Fmoc-protected lysine, and compound **2,** with an Fmoc-protected arginine, were synthesized via the amide coupling of MeRho with the respective Fmoc-protected amino acids (Fmoc-Lys(Boc)-OH and Fmoc-Arg(Pbf)-OH) using the amide coupling reagents EEDQ and EDC/DMAP, respectively. Compounds **5** and **6** were synthesized by deprotection of Fmoc using piperidine followed by amide coupling with Fmoc-Gly-OH using DIC and HOBt. Repetition of Fmoc deprotection and amide coupling with Ac-Leu-OH under the same reaction conditions provided compounds **9** and **10**. Finally, acid-catalyzed (TFA) deprotection yielded the fluorogenic tripeptide substrates MeRhO-Lys-Gly-Leu(Ac) **11** and MeRhO-Arg-Gly-Leu(Ac) **12**.

### 2.2. Development of a Fluorogenic Assay for Determination of the Extent and Chemoselectivity of Lysine and Arginine Chemical Modification

#### 2.2.1. Buffer Selection

Different buffers have unique and pronounced effects on enzymatic reactions. At pH values substantially higher than physiological pH (pH 7.4), most enzymes will undergo denaturation or alkaline hydrolysis [28]. It is, thus, necessary to maintain an optimum pH at which the side chain amino group of the amino acid will be deprotonated with good nucleophilicity that permits a reasonable reaction with the blocking agent [8]. With respect to its proteolytic activity, enzyme trypsin has an optimal pH range of between 7.5 and 8.5 [29]. Buffers used for chemical modification of amino acid side chains should ideally be free of the primary amino group to avoid undesirable side reactions. On the basis of the aforementioned considerations, we performed a buffer selection study on 1 µM of the MeRho-Lys-Gly-Leu(Ac) peptide substrate using the three buffers HEPES (50 mM), PBS (10 mM), and sodium borate (0.1 mM) at pH 8.3, with the aim of identifying a buffer facilitating complete cleavage, thereby giving rise to high fluorescence intensity (MeRho: λex/λem 476/516) (Figure 3). We accordingly found that all buffers worked well, as evidenced by an increase in fluorescence over time when the peptide substrate was reacted with trypsin. However, given that compared with the other two buffers, the use of HEPES buffer gave rise to a higher fluorescence at 30 min, we selected this as the reaction buffer for further experiments.

#### 2.2.2. Stability of Fluorogenic Peptide Substrates

The stability of a given fluorogenic peptide substrate is a particularly important consideration with respect to the buffer conditions required for fluorogenic assays, and thus we assessed the stability of the peptide linkages in the fluorogenic peptide substrates by measuring the fluorescence intensity under different temperature and pH conditions (Figure 4). We accordingly found that both fluorogenic peptide substrates showed stable fluorescence emissions in the pH and temperature ranges between 2 and 10 and 25 °C and 45 °C, respectively, thereby indicating that these substrates could be used for the development of fluorogenic assays with wide pH and temperature ranges.

#### 2.2.3. Kinetic Study of Trypsin Activity on Fluorogenic Tripeptide Substrates

We went on to determine whether the newly synthesized fluorogenic peptide substrates were sensitive to trypsin concentration, for which we measured the fluorescence produced by 1 μM of peptide substrate (MeRho: 476 nm excitation and 516 nm emission) on reaction with different concentrations of the trypsin enzyme (Figure 5). Initially, the kinetic study of proteolytic activity was performed using trypsin concentrations ranging from 0.025 to 25 μg/mL (Figure 5A,C for MeRho-Lys-Gly-Leu(Ac) and MeRho-Arg-Gly-Leu(Ac), respectively), and we found that trypsin concentrations of 0.25 and 1 μg/mL produced a kinetically stable fluorescence. Thereafter, we sought to determine the optimum trypsin concentration within the range of 0.25 to 1 μg/mL required for proteolysis (Figure 5B,D for MeRho-Lys-Gly-Leu(Ac) and MeRho-Arg-Gly-Leu(Ac), respectively). The concentration of MeRho released by proteolysis of the fluorogenic peptide substrates was also calculated from the calibration curve of MeRho standards (Figure 5E,F). We accordingly established that application of trypsin at a concentration of 1 µg/mL produced the best results with the highest fluorescence values, corresponding to the release of 61% and 94% of MeRho from MeRho-Lys-Gly-Leu(Ac) and MeRho-Arg-Gly-Leu(Ac), respectively, and accordingly used this trypsin concentration in further assays.

#### 2.2.4. Determination of the Extent and Chemoselectivity of Chemical Modification of Lysine and Arginine Using a Fluorogenic Assay

We subsequently investigated whether our novel fluorogenic peptide substrates are practically applicable for determining the extent and chemoselectivity of lysine and arginine chemical modification. For this purpose, we selected lysine blockers (anhydrides and cyclic anhydride derivatives) and arginine blockers (α-dicarbonyl derivatives) as blocking agents for side-chain modification [8,17]. These blockers were screened to assess their capacity to chemoselectively modify an amino acid (AA1) in the fluorogenic peptide substrates. Chemical modification of AA1 modulated the proteolytic activity of trypsin toward the substrate, thereby reducing fluorescence, depending on the concentration of the blocker used in the fluorogenic assay.

Different concentrations of the selected lysine blockers, namely, anhydrides (acetic anhydride, benzoic anhydride, diethylpyrocarbonate, *p*-toluene sulfonic anhydride) and cyclic anhydrides (maleic anhydride, citraconic anhydride, and phthalic anhydride) were reacted with both fluorogenic substrates for 30 min, after which we assessed the effects on the proteolytic activity of trypsin for 30 min. Having obtained semilogarithmic plots of concentration–response curves (see Appendix A to Appendix A in ESI), we used these to calculate IC_50_ values to quantify the extent of chemical modification. The chemoselectively of chemical modification for either lysine or arginine was assessed based on the IC_50_ ratios IC_50_arginine/IC_50_lysine and IC_50_lysine/IC_50_arginine, (Table 1 and Table 2). In this regard, lysine contains an ε-amino group (pKa ~ 10.5) [30] as a side chain, whereas arginine contains a guanidino group (pKa 12–13.7) [31,32], and on the basis of the reactivities of these side chains at slightly basic pH, these amino acids are chemically modified by anhydrides and glyoxals, respectively [8]. The deprotonated primary amine in the side chain of lysine reacts rapidly and specifically with anhydrides via a nucleophilic acyl substitution reaction to form amide bonds [8], whereas the addition of α-dicarbonyl to arginine results in the formation of a hydrolytically unstable imidazolidine, which is stabilized by the addition of a one more mole of glyoxal [8].

In the present study, we found that both anhydrides and cyclic anhydrides showed chemoselective blocking activity for lysine compared with arginine, whereas sulfonic anhydrides proved to be ineffective in blocking the activity of either of the two amino acids (Table 1). Among these anhydrides, benzoic anhydride was established to have the lowest IC_50_ value of 4.90 µM toward lysine, the chemoselectivity of which was 17 times higher than that for arginine. Interestingly, cyclic anhydrides were found to be best for chemoselective lysine chemical modification, with no detectable modification of arginine.

We similarly investigated the reactivity of arginine-specific blockers toward the two fluorogenic peptide substrates (Table 2), and accordingly found that α-dicarbonyl compounds such as phenyl glyoxal derivatives were highly arginine-selective in our fluorogenic assay, which is consistent with the findings of previous studies [15]. Substituted phenylglyoxal derivatives also showed high arginine selectivity and, with the nature and position of the substituent in phenylglyoxal determining the extent of modification. Compared with *ortho*-substitution (2-CF_3_), the electron-withdrawing group (4-CF_3_ or 4-NO_2_) at *para* position favored arginine modification, whereas an electron-donating group (4-OCH_3_), at the *para* position showed relatively less interaction with arginine. However, we found that benzaldehyde with a single aldehyde group does not interact with arginine, thereby indicating that the dicarbonyl moiety plays a significant role in the interaction with arginine, as reported previously. Surprisingly, we observed that the introduction of an additional aromatic ring (naphthyl) showed the most potent inhibition for arginine, although chemoselectivity was lost. However, vicinal diketones (1,2-cyclohexadione and 2,3-butadione), previously reported to be arginine-specific blockers, were found to show no reactivity with arginine in our fluorogenic assay, which could conceivably be attributed to a reversible reaction resulting in the formation of unstable *cis*-diol and dihydroxyimidazoline, and needs to be addressed separately [8,33].

## 3. Materials and Methods

### 3.1. Materials and Instrumentation

All starting materials and reagents were purchased from Sigma-Aldrich Chemical Co. (St. Louis, MO, USA); Tokyo Chemical Industries (Tokyo, Japan); Daejung Chemicals (Siheung-si, Korea) and Alfa Aesar (Ward Hill, MA, USA), and were used without any further purification. Solvents were purified using a PureSolv Micro Multi Unit solvent purification system obtained from Inert Technology (Amesbury, MA, USA) and were used under a dry nitrogen atmosphere. The progress of reactions was assessed by thin-layer chromatography on silica gel plates (Kiesegel 60F_254_; Merck; Darmstadt, Germany), and the synthesized compounds were purified by flash column chromatography using silica gel (ZEOprep 60; 40–63 μm; Zeochem, Louisville, KY, USA). ^1^H-NMR and ^13^C spectra were measured with a JEOL JNM-ECZ400s/L1 (400 MHz) spectrometer (Tokyo, Japan), using CDCl_3_ or DMSO-*d*_6_ as the NMR solvent (Cambridge Isotope Laboratories, Tewksbury, MA, USA). ^1^H-NMR chemical shifts are expressed in terms of parts per million (ppm) based on the chemical shift of tetramethylsilane (δ = 0 ppm) in CDCl_3_ as an internal standard. The chemical shifts in ^13^C-NMR are reported in ppm relative to the centerline of the triplet at 77.0 ppm observed for CDCl_3_ or 39.5 ppm for DMSO-*d*_6_. The coupling constant J in ^1^H-NMR is reported in hertz (Hz). Fluorogenic assays were performed using a Synergy™ H1 microplate reader from BioTek Instruments (Winooski, VT, USA). Trypsin from porcine pancreas was purchased from Sigma-Aldrich Chemical Co. (St. Louis, MO, USA). The enzyme solution was freshly prepared prior to performing assays by dissolving lyophilized trypsin powder in assay buffer (50 mM HEPES, pH 8.3). Stock solutions of the fluorogenic peptide substrate and amino acid blockers were prepared in DMSO.

### 3.2. General Synthetic Procedure

#### 3.2.1. General Procedure A: Amide Coupling

To a solution of aniline or amine (1.0 eq.) and AA (1.2 eq.) in CH_2_Cl_2_, we added DIC (1.2 eq.) and HOBt (1.2 eq.). The reaction mixture was stirred at room temperature for 2 h, and on completion of the reaction, the reaction solvent was evaporated. The resulting solid was purified by column chromatography to obtain the desired product.

#### 3.2.2. General Procedure B: Fmoc Deprotection

To a solution of Fmoc-protected compound (1.0 eq.), we added piperidine (1.2 eq.) in anhydrous CH_3_CN (395 eq.). The reaction mixture was stirred at room temperature for 2 h, and on completion of the reaction, the reaction solvent was evaporated. The resulting residue was purified by column chromatography to obtain the desired Fmoc-deprotected product.

#### 3.2.3. General Procedure C: Acid-Labile Group Deprotection (Boc/Pbf)

To a solution of Boc/Pbf-protected compound (1.0 eq.), we added TFA (10% solvent) in anhydrous CH_2_Cl_2_. The reaction mixture was stirred at room temperature for 2 h, and on completion of the reaction, the pH of the reaction mixture was increased with 1N NaOH. The compound was extracted using an organic solvent mixture (DCM: MeOH, 90:10), and the organic layer was dried over Na_2_SO_4_ and filtered, with the resulting filtrate being evaporated under vacuum. The resulting residue was purified by column chromatography to obtain the desired fluorogenic peptide substrate.

### 3.3. Buffer Selection

Buffer selection was performed by incubating 1 µM of the fluorogenic peptide substrate (MeRho-Lys-Gly-Leu(Ac)) in different buffers (HEPES (50 mM, pH 8.3), PBS (10 mM, pH 8.3), and sodium borate (0.1 mM, pH 8.3)) at 30 °C in presence (2 µg/mL) or absence of trypsin and recording the fluorescence (λex/λem 476/516) in each buffer in a 96-well microplate using a Synergy H1 reader.

### 3.4. Thermal and pH Stability of Fluorogenic Peptide Substrates

We performed a temperature-dependent assay by incubating 1 µM of the fluorogenic peptide substrate in HEPES buffer (50 mM, pH 8.3) at different temperatures (25, 28, 31, 34, 37, 43, and 45 °C) for 20 min, and fluorescence was recorded (λex/λem 476/516) at each temperature. An assessment of pH dependence was performed by incubating 1 µM of the fluorogenic peptide substrate in a range of pH buffer solutions (pH 2 to 13, at 30 °C) and recording the fluorescence (λex/λem 476/516) at each pH in a 96-well microplate using a Synergy H1 reader.

### 3.5. Sensitivity of Fluorogenic Peptide Substrates to Trypsin Proteolysis

The sensitivity of the fluorogenic peptide substrate to the proteolytic activity of trypsin was assessed by incubating 1 µM of substrate in different concentrations of trypsin in HEPES buffer (50 mM, pH 8.3) at 30 °C and recording the fluorescence spectra (λex/λem 476/516) kinetically at each trypsin concentration in a 96-well microplate using a Synergy H1 reader.

### 3.6. Determination of Concentration of Fluorophore Released from Fluorogenic Peptide Substrate

Fluorescence of a series of standard solutions of MeRho with concentration range of 0.005, 0.01, 0.02, 0.05, 0.1, 0.2, 0.5, and 1.0 μM was measured in a 96-well microplate using a Synergy H1 reader at λex/λem 476/516. A first order straight line was fitted in graph with fluorescence intensity (*y*-axis) and concentration μM (*x*-axis). The concentration of fluorophore released from substrate on trypsin proteolysis was calculated from the slope of the calibration graph (Figure 5E,F).

### 3.7. In Vitro Fluorogenic Assay

All spectroscopic readings were recorded with a BioTek Synergy^TM^ H1 instrument using a 96-well microplate. The proteolytic reaction was performed in a total volume of 200 µL with the addition of 100 µL of HEPES (50 mM, pH 8.3), 10 µL (1 µM) of fluorogenic peptide substrate stock solution (20 µM in DMSO), and 2 µL (final concentration of blocker 0.01, 0.1, 1, 10, 100, 1000, and 10000 µM) of amino acid blocker stock solution (0.001, 0.01, 0.1, 1, 10, 100, and 1000 mM), with the final volume being adjusted to 190 µL using HEPES buffer (50 mM, pH 8.3). The plates were incubated at 30 °C for 30 min with continuous shaking. Thereafter, 10 µL (1 µg) of trypsin solution (20 µg/1000 µL in HEPES, 50 mM, pH 8.3) was added to the assay mixture, and the plate was again incubated at 30 °C for 30 min with continuous shaking. For preparation of a kinetic graph, emission spectra were recorded at λex/λem 476/516 with respect to time. In the case of cyclic anhydrides (i.e., citraconic anhydride, maleic anhydride, and phthalic anhydride), 2 µL (final concentrations of NaOH (0.02, 0.2, 2, 20, 200, 2000, and 20,000 µM) of NaOH stock solution (0.002, 0.02, 0.2, 2, 20, 200, and 2000 mM) was also added to maintain the pH of the assay mixture. Semilogarithmic plots were constructed using values at 30 min after proteolysis to calculate the IC_50_ values of amino acid blocking agents.

## 4. Conclusions

In conclusion, we developed a highly efficient innovative fluorogenic assay, based on a fluorogenic peptide substrate containing a lysine and arginine at proteolytic site, to determine the extent and chemoselectivity of chemical modification of amino acids lysine and arginine. We designed two fluorogenic tripeptide substrates, MeRho-Lys-Gly-Leu(Ac) and MeRho-Arg-Gly-Leu(Ac) comprising a MeRho fluorophore conjugated to lysine or arginine, a site for selective chemical modification, and demonstrated that these substrates exhibited turn-on fluorescence via release of the fluorophore following cleavage by a trypsin. Conversely, the chemoselective chemical modification of amino acids lysine or arginine leads to a decrease in fluorescence based on the extent of modification of the proteolytic reaction. Using this assay, we also demonstrated that changes in the electronic factor and position of the substituent in the blocking agents affect the extent and chemoselectivity of amino acid modification. Most commonly used blocking agents including anhydrides or 1,2 dicarbonyl derivatives to modify amino acids lysine or arginine during peptide or protein modification was not well characterized, lacking the detailed study about the extent and chemoselectivity for a specific amino acid. The findings of this study indicate that the novel fluorogenic assay will have potential utility in the characterization of blocking agents and used them more efficiently during peptide or protein modification. In addition, it is possible to design and development of new more efficient and highly selective amino acid blockers for lysine or arginine, by performing structure–activity relationship analyses that assess the substituent effect of amino acid blockers as performed for phenyl glyoxal in this study. The amino acid blockers identified using fluorogenic assay in this study can be applied further for peptide or protein modification to achieve desired objective.

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
