# Peer review of "A Fluorogenic Assay: Analysis of Chemical Modification of Lysine and Arginine to Control Proteolytic Activity of Trypsin"

_molecules, 2021, doi:10.3390/molecules26071975_

Round 1

Reviewer 1 Report

New methods for detecting and measuring the activity of proteases, as well as the development of molecular biological constructs controlled by proteases, are extremely important, both in fundamental biology and in medicine. In this article, the authors synthesized a fluorophore that is not optically active when a tripeptide is attached. When the tripeptide is cut off from the fluorophore, the fluorophore becomes optically active. In general, the scheme is quite clear. The novelty of the work is ensured by a new optically active connection. The results of the work are presented clearly and understandably. Despite the overall high assessment of the work, I have a number of non-essential comments.
1. Trypsin has a much narrower optimum of pH values, approximately 7.8-8.0.
2. The authors use pH 8.3 buffers (PBS, HEPES, borate). Why? This is not a fluorophore optimal not trypsin optimum. By the way, our research (with a similar optically active compound) showed, that the most effective may be a carbonate buffer.
3. In fig. 4, the dimension of the Y axis should be reduced by at least one order of magnitude. The experimental curves merge with the X-axis, nothing can be made out.

Reviewer 2 Report

The manuscript by More et al. describes the development of a fluorogenic assay to assess the extent and site selectivity of chemical modifications of amino acids. The assay's validity is demonstrated using a string of two specific sequences of amino acids and a common proteolytic enzyme. The method has potential application in diverse fields.

The paper is overall very well written, and the results well presented. The introduction clearly enunciates the problem and the methods described with sufficient detail. 

Specific comments to the manuscript are:

- the paper would benefit from testing the assay in more complex substrates such as more complex peptides or proteins; while this might be out of the scope of the present manuscript and explored in another manuscript, the present manuscript would benefit from a discussion/comment on the feasibility of using this assay in biological samples.

- figures 2 and 3 would benefit from more detailed figure legends; for instance, in figure 3, what is “-ve”; one assumes is the absence of trypsin (this aspect is also true for figure 4).

Reviewer 3 Report

In this manuscript, the authors have developed a fluorogenic assay to determine the extent of site-selective modification of amino acids. They have further used this assay to identify structural and electronic factors that influence the efficacy and selectivity of the amino acid blocking reagents. However, the following points must be addressed prior to acceptance of the manuscript.

  1. The authors claim that this assay can be used to screen for new blocking groups. For this to be true they should demonstrate that the findings from their assay are translatable to either proteins or longer peptides by an orthogonal technique. Does the best performing blocking group identified from this fluorogenic assay perform similarly when a protein or longer peptide is used?
  2. Does the context of the lysine or arginine residues matter in terms of blocking efficacy? In other words, does it matter what amino acids are around the lysine or arginine residues targeted? The authors should test this using substrates where the lysine or arginine residues have other neighboring amino acids. Also, if there are multiple lysine or arginine residues, is there a preferential modification of certain lysine/arginine residues?
  3. Figure 4B: There is a slight increase observed at higher temperatures. Is this difference statistically significant compared to the 25°C value?
  4. Section 2.2.3: “We accordingly established that application of trypsin at a concentration of 1μg/mL produced the best results with the highest fluorescence values, corresponding to the release of 61% and 94% of MeRho from MeRho-Lys-Gly-Leu(Ac) and MeRho-Arg-Gly-Leu(Ac), respectively, and accordingly used this trypsin concentration in further as-says”. In materials and methods provide details of the calculations for %MeRho released.
  5. Figure 5F: Why is the concentration of fluorophore released from 0.45ug of trypsin less than that released from 0.25ug trypsin for both peptides? Is this difference statistically significant?
  6. Supplementary fig S11: 4-nitro Phenyl glyoxal structure is missing.

Round 2

Reviewer 3 Report

The revised manuscript adequately addresses the previously raised concerns and is now suitable for publication.